# Qualitative and Quantitative Analysis of Secondary Metabolites in Morphological Parts of Paulownia Clon In Vitro 112^®^ and Their Anticoagulant Properties in Whole Human Blood

**DOI:** 10.3390/molecules27030980

**Published:** 2022-02-01

**Authors:** Anna Stochmal, Barbara Moniuszko-Szajwaj, Jerzy Zuchowski, Łukasz Pecio, Bogdan Kontek, Malgorzata Szumacher-Strabel, Beata Olas, Adam Cieslak

**Affiliations:** 1Department of Biochemistry and Crop Quality, Institute of Soil Science and Plant Cultivation—State Research Institute, Czartoryskich 8, 24-100 Puławy, Poland; asf@iung.pulawy.pl (A.S.); bszajwaj@iung.pulawy.pl (B.M.-S.); jzuchowski@iung.pulawy.pl (J.Z.); lpecio@iung.pulawy.pl (Ł.P.); 2Department of General Biochemistry, Faculty of Biology and Environmental Protection, University of Lodz, 90-236 Łódź, Poland; bogdan.kontek@biol.uni.lodz.pl; 3Department of Animal Nutrition, Poznań University of Life Sciences, Wołyńska 33, 60-637 Poznań, Poland; mstrabel@joy.up.poznan.pl (M.S.-S.); adamck@joy.up.poznan.pl (A.C.)

**Keywords:** anticoagulant activity, Paulownia Oxytree, secondary metabolites, morphological parts, T-TAS

## Abstract

It is not easy to find data in the scientific literature on the quantitative content of individual phytochemicals. It is possible to find groups of compounds and even individual compounds rather easily, but it is not known what their concentration is in cultivated or wild plants. Therefore, the subject of this study was to determine the content of individual compounds in the new Paulownia species, Oxytree, developed in a biotechnology laboratory in 2008 at La Mancha University in Spain. Six secondary metabolites were isolated, and their chemical structure was confirmed by spectral methods. An analytical method was developed, which was then used to determine the content of individual compounds in leaves, twigs, flowers and fruits of Paulownia Clon in Vitro 112^®^. No flavonoids were found in twigs and fruits of Oxytree, while the highest phenylethanoid glycosides were found in twigs. In this study, we also focused on biological properties (anticoagulant or procoagulant) of extract and four fractions (A–D) of different chemical composition from Paulownia Clon in Vitro 112 leaves using whole human blood. These properties were determined based on the thrombus-formation analysis system (T-TAS), which imitates in vivo conditions to assess whole blood thrombogenecity. We observed that three fractions (A, C and D) from leaves decrease AUC_10_ measured by T-TAS. In addition, fraction D rich in triterpenoids showed the strongest anticoagulant activity. However, in order to clarify the exact mechanism of action of the active substances present in this plant, studies closer to physiological conditions, i.e., in vivo studies, should be performed, which will also allow to determine the effects of their long-term effects.

## 1. Introduction

Paulownia is a genus of plants in the Paulowniaceae family, formerly classified as Scrophulariaceae. The plant was named in honour of the Queen of the Netherlands, Anna Pavlovna Romanov, who financed the trip to China from which this tree was first brought. The species of this genus include: *Paulownia catalpifolia* T. Gong ex D.Y. Hong, *Paulownia elongata* S.Y. Hu, *Paulownia fargesii* Franch., *Paulownia fortunei* (Seem.) Hemsl., *Paulownia kawakamii* T. Itô, *Paulownia taiwaniana* T.W. Hu and H.J. Chang, *Paulownia tomentosa* Steud. Paulownia Clon in Vitro 112, a hybrid of two species (*Paulownia fortunei* and *Paulownia elongata*), known under the trade name Oxytree, has been popularized as a tree with exceptionally strong biomass growth, in the first two years of growth, with huge leaves producing ten times more oxygen than any other tree. Oxytree wood is half the weight of other hardwoods, and once dried it does not gain moisture. Growing Oxytree is a perpetual motion, because a tree once planted is cut after six years, when it reaches 16 m in height and 35 cm in diameter, and then grows back from the trunk and grows so quickly that after four years it is ready to be cut again—such a process is repeated four more times. Paulownia Clon in Vitro112, unlike other Paulownia species, does not propagate by seeds or root cuttings, but only in vitro.

The species tested for the content of secondary metabolites is *P. tomentosa var. tomentosa*. The authors of these studies found secondary metabolites as shown in Table 1.

*Paulownia tomentosa* is a rich source of secondary metabolites of groups such as: Non-prenylated flavonoid aglycones, C-prenylated and C-geranylated flavonoids, flavonoid glycosides, iridoids, phytosterols and phenolic acids. Biologically active compounds are used as traditional Chinese herbal remedies, and also influence modern healthcare.

Flavonoids have a potential role as anti-cancer compounds for use in cervical and breast [25], hepatoma, leukemia [3], osteosarcoma [26], gastric cancer [27], colon adenocarcinoma [28] or prostate [29] cancer cell lines. Their antioxidant properties explain the cardioprotective effect [30,31] and neuroprotective action [32]. They have been found to exhibit antibacterial properties [21,33,34] and antiviral [35] biological activity against various pathogens.

Iridoid–catalpol shows radioprotective activity [26] and neuroprotective [36] and cardioprotective effects on the heart muscle (Human); it also increases glucose utilization by increased secretion of β-endorphin from the adrenal gland [37]. Aucubin had an antioxidant and protective effect on the pancreas and can alleviate obesity-induced atherosclerosis [38].

Phytosterols are potentially useful for the treatment of Alzheimer’s disease [39]. They show a cytotoxic effect against K562 and K562/ADR human chronic myeloid leukemia, human HL60 and HL60/ADR acute myeloid leukemia cancer cells and neoplastic cell human colon cancer cell lines SW480 and SW620 [40]. They have anti-inflammatory [41,42], antimalarial [43] and anti-diabetic effects [44], reduce blood pressure and are useful in the treatment of traumatic cerebral ischemia [45].

In the present study, a targeted identification and quantification of the secondary metabolites, present in Paulownia Clon in Vitro 112, was achieved by UPLC-MS/MS and UPLC-DAD. Our earlier results demonstrated that extract and four fractions (A–D) from leaves of Paulownia Clon in Vitro 112 have antioxidant and anti-platelet potential in vitro. In these studies, we used human plasma and washed human blood platelets [46,47]. Therefore, in this study, we also focused on biological properties (anticoagulant or procoagulant) of extract and four fractions (A–D) of different chemical composition from Paulownia Clon in Vitro 112 leaves using whole human blood. These properties were determined based on the thrombus-formation analysis system (T-TAS), which imitates in vivo conditions to assess whole blood thrombogenecity.

## 2. Materials and Methods

### 2.1. Chemicals

Acetonitrile (LC-MS grade) and formic acid (LC-MS grade) were obtained from Merck (Darmstadt, Germany). Verbascoside was obtained from HWI Analytik (Ruelzheim, Germany), rutin from PhytoLab (Vestenbergsgreuth, Germany). Catalpol, maslinic acid and dimethyl sulfoxide (DMSO) from Sigma Aldrich (St. Louis, MO, USA). The PL-chip and other equipment needed for the T-TAS were purchased from Bionicum (Lodz, Poland).

### 2.2. Plant Material

For the extraction of compounds, the leaves collected on the private plantation were used in the first year after its establishment, early autumn (September), on the fluvisol developed on Vistula river deposits, class I, in the village Łęka, Lublin Province, Poland (21°54′ N, 51°27′ E).

The owner—Anna Stochmal—has a certificate for the cultivation of trees. She bought seedlings from the official representative in Poland: Oxytree S.A., ul. Życzliwa 25/2, 53-030 Wrocław.

Oxytree were identified by Sales Director Dariusz Tyma.

Qualitative and quantitative analyses of the morphological parts—leaves, twigs, flowers and fruits—were performed on plant material collected on a 5-year-old plantation located on sandy land, class IV, Września-Wielkopolska-Poland (52°20′40.7″ N 17°36′28.6″ E).

Plants after harvest were frozen and freeze-dried (Gamma 2-16 LSC, Christ, Germany). Reference samples are stored at the Institute of Soil Science and Plant Cultivation.

The sample voucher for this material has been deposited in the institute’s collection under the deposit number 22/2017/IUNG.

### 2.3. Extraction and Isolation of Compounds

A sequential extraction process was used to prepare an extract of Paulownia leaves. The ground plant material was extracted with 5% methanol (*v*/*v*) in an ultrasonic bath at room temperature for 30 min and macerated with a magnetic stirrer at room temperature for half an hour. The contents were centrifuged at 4000 G for 10 min and filtered. The pellet was re-extracted with 30% methanol (*v*/*v*) and the third extraction was done with 70% methanol. The supernatants were combined and concentrated under reduced pressure and lyophilized. The extraction yield was 41.7%.

The crude methanol extract was purified stepwise using various chromatographic methods. First, the extract was applied to a pre-conditioned RP-C18 column (80 × 70 mm, 140 µm; Cosmosil C18-PREP; Nacalai Tesque, Inc., Kyoto, Japan), which was then washed with 1% methanol (*v*/*v*) to remove sugars. Active metabolites were eluted with 80% methanol (*v*/*v*) to give fraction A. The yield at this stage was 44.1%.

In the next step fraction A was separated by flash chromatography on a reverse phase column (140 × 12 mm, 40 µm; Cosmosil C18-PREP, Nacalai Tesque, Inc., Kyoto, Japan) connected to a Gilson HPLC apparatus. A linear gradient of an aqueous acetonitrile solution (2–30% *v*/*v*) containing 0.1% formic acid over 140 min was used as the mobile phase at a flow rate of 8 mL min^−1^ at room temperature. Subsequently, from fraction A, sub-fractions B, C and D were obtained, which, respectively, accounted for 23.4%, 33.6% and 36.3% of fraction A.

Isolation of compounds from subfractions was made on column Atlantis Prep T3, C18 5 µm, 10 × 250 mm (Waters) by preparation chromatography using isocratically different concentration of methanol. From subfraction B two compounds were purified, from C three compounds, while from D one compound. A more detailed description of preparation and the extract and four fractions (A–D) from leaves can be found in the paper of Adach et al. [46].

### 2.4. NMR Spectroscopy

The 1D and 2D NMR spectra of isolated compounds, including ^1^H-, ^13^C-DEPTQ (distortionless enhancement by polarization transfer with retention of quaternaries), selective TOCSY (total correlation spectroscopy), selective ROESY (rotation frame nuclear overhauser effect spectroscopy), ^1^H-^1^H-DQF-COSY (double-quantum filtered correlation spectroscopy), ^1^H-^13^C-HSQC (heteronuclear single quantum coherence) and ^1^H-^13^C-HMBC (heteronuclear multiple bond coherence) were recorded using an Avance III HD Ascend 500 MHz spectrometer (Bruker BioSpin, Rheinstetten, Germany) in deuterated methanol (MeOH-*d*_4_) at 30 °C. NMR spectra were calibrated to the signal of residual solvent: δ 3.31 for ^1^H and 49.0 for ^13^C [48,49].

### 2.5. Analytical Method UHPLC-ESI-MS/MS

Extracts and fractions of Paulownia leaves, flowers, fruit and twigs were prepared by the same method described above (Section 2.3) and next were analyzed by UHPLC-ESI-MS/MS, using a Thermo Ultimate 3000RS (Thermo Fischer Scientific, MS, Carlsbad, CA, USA) UHPLC system, equipped with a charged aerosol detector (CAD) and a diode array detector, and hyphenated with a Bruker Impact II (Bruker Daltonics GmbH, Bremen, Germany) Q-TOF mass spectrometer. Samples were chromatographed on an ACQUITY HSS T3 column (2.1 × 150 mm, 1.8 µm; Waters, MA, USA) maintained at 55 °C; the injection volume was 2.0 µL. A 22 min gradient elution program was applied, from 2 to 45% of solvent B (0.1% formic acid in acetonitrile) in solvent A (0.1% formic acid in Milli-Q water, Sartorius, Gottingen, Germany); the flow rate was 0.500 mL min^−1^. The applied MS settings were similar to those described previously [50,51], with small modifications. UHPLC-MS/MS analysis was performed in negative ion mode, the scanning range was from *m/z* 80 to 2000. The following settings were applied: Capillary voltage was 3 kV, dry gas flow was 6 L min^−1^, dry gas temperature was 200 °C, nebulizer pressure was 0.7 bar, collision RF was 750 V, transfer time was 100 ms, prepulse storage time was 10 ms. Two precursor ions of intensity over 2000 counts were fragmented in each scan. Depending on the *m/z* of a fragmented ion, the value of collision energy was set automatically in the range from 2.5 to 80 eV. The internal calibration of the acquired data was performed with sodium formate, introduced to the ion source via a 20 µL loop at the beginning of a separation. Constituents of the extracts were determined by external calibration, using authentic standards when possible. The UV detection at λ = 330 nm was used for quantitation and semi-quantitation of phenolic compounds. The content of verbascoside was determined on the basis of the following standard curve (y = 4.90599x − 1.97683; R^2^ = 0.9999). The same standard curve was also used for semi-quantitation of other phenylethanoids, and hydroxycinnamic acid derivatives. Flavone glycosides were determined on the basis of a standard curve of apigenin-7-*O*-glucuronopyranosyl-(1→2)-glucurono-pyranoside (y = 6.89810x − 19.60755; R^2^ = 0.9997). Apart from phenolics, two iridoids were additionally quantified. 7-hydroxytomentoside was determined on the basis of UV chromatograms (λ = 248 nm), with a standard curve y = 4.02230x − 1.07988; R^2^ = 0.9995. Extracted ion chromatograms (EIC) were used for quantitative analyses of catalpol, on the basis of the following standard curve y = −41.329553x^2^ + 17210.064761x + 12019.481295; R^2^ = 0.9972).

### 2.6. Preparation of Stock Solutions (Extract and Four Fractions (A–D) from Leaves) for Bioassay—T-TAS

The extract and the four fractions (A–D) from leaves were dissolved in 1 mL 50% DMSO (as a universal solvent for many different plant substances), giving final concentrations of 5 μL/mL and 50 μL/mL (in blood samples). The final concentration of DMSO in the tested blood samples was below 0.05% (*v*/*v*). The addition of a low concentration of DMSO to human blood has no effect on coagulation parameters (data not presented).

### 2.7. The Samples of Blood

Fresh human blood was collected in the L. Rydygier hospital in Lodz, Poland. All donors were healthy volunteers; none were smokers or reported taking drugs. Blood was collected in tubes with CPDA anticoagulant (citrate/phosphate/dextrose/adenine; 8.5:1; *v*/*v*; blood/CPDA). The analysis of blood samples was performed according to the guidelines of the Helsinki Declaration for Human Research. The blood used in T-TAS assays was incubated (30 min, at 37 °C) with extract and four fractions (A–D) from leaves at final concentrations of 5 and 50 μg/mL. The experiments were conducted with the consent of the Bioethics Committee at the University of Łódź (number 11/KBBN-UŁ/I/2019).

### 2.8. Total Thrombus-Formation Analysis System (T-TAS)

The plate plug formation process was determined using a real-time hydrodynamic model. Whole blood (400 µL) was incubated with the tested extract and four fractions (A–D) (30 min, 37 °C). Next, 350 µL of blood was drawn for analysis. The results were obtained as AUC_10_ (Area Under the Curve) using a PL chip. The AUC_10_ parameter—that is an area under the pressure curve from the start of the test to a time of 10 min—was studied. This parameter determines the growth, intensity, and stability of thrombus formation. A more detailed description of this method can be found in Hosokawa et al. [52].

### 2.9. Data Analysis

Statistical analysis was performed using Statistica 10 (StatSoft 13.3, TIBCO Software Inc., Palo Alto, CA, USA). Normal distribution of data was checked by normal probability plots and the homogeneity of variance by the Brown–Forsythe test. Differences within and between groups were assessed using one-way ANOVA followed by a multicomparison Duncan’s test. Results are presented as means ± SD. Significance was considered at *p* ≤ 0.05. Dixon’s *Q*-test was used to eliminate uncertain data.

## 3. Results

The LC-QTOF-MS analysis of the extract from Paulownia Clon in Vitro 112 leaves, grown in Łęka, shows that the main secondary metabolite was the verbascoside (acteoside), which belongs to the phenylethanoid glycosides. It contained also other phenylethanoides. Iridoids such as catalpol, 7-hydroxytomentoside, as well as flavonoids: Glycosides of luteolin, and apigenin, were also found (Figure 1).

Multi-step chromatographic separation led to the isolation of several polyphenolic constituents—derivatives of a disaccharide α-l-rhamnopyranosyl(1→3)-d-glucopyranose—verbascoside (1), verbascoside (2) and cistanoside F (3); and a glycosylated flavone clerodendrin—apigenin 7-*O*-[β-d-glucuronopyranosyl(1→2)-β-d-glucuronopyranoside] (4); but also two iridoid glycosides—catalpol (5) (as a mixture) and 7-hydroxytomentoside (6) (Figure 2). Their identity was confirmed by means of accurate mass measurements, MS/MS fragmentation patterns, ultraviolet-visible (UV-Vis) spectra, 1D and 2D NMR (nuclear magnetic resonance) spectroscopy, and by comparison with existing literature data. The data are in the Appendix A (Appendix A).

The UHPLC-ESI-MS/MS method was used to determine the content of individual compounds in leaves, twigs, flowers and fruits of Paulownia Clon in Vitro from Września. All verbascoside derivatives, including cistanoside F, were determined as equivalent of verbascoside, while all flavonoids were determined to be apigenin-7-*O*-glucuronopyranosyl(1→2)-glucuronopyranoside equivalent (Table 1). The concentrations of the individual compounds were determined according to the calibration curves of standards (Table 1), and classes of the individual compounds in Table 2 were created by summing up the concentrations of individual compounds.

The results demonstrated alterations in AUC_10_ measured by T-TAS in the presence of all tested plant preparations from leaves (Figure 3). Tested extract and fractions (A–D) from leaves (at concentrations 5 and 50 µg/mL) decreased AUC_10_ relative to control, but these changes were not always statistically significant (Figure 3). However, for three fractions (A, C and D; at the highest used concentration—50 µg/mL) AUC_10_ decreased in a statistically significant manner. In addition, fraction D (50 µg/mL) had the strongest effect on this parameter (Figure 3). Figure 4 demonstrates selected diagrams analysed by T-TAS for tested extract (50 µg/mL) and four used fractions (A–D, 50 µg/mL) in whole blood samples. We have calculated AUC_10_ for each diagram.

## 4. Discussion

The most widespread Oxytree compound is verbascoside. Verbascoside (acteoside) belongs to phenylpropanoid glycosides—a group of phenolic compounds widespread among higher plants, especially from the families: *Araliaceae*, *Bignoniaceae*, *Crassulaceae*, *Labiateae*, *Oleaceae*, *Plantaginaceae*, *Polygonaceae*, *Scrophulariaceae*, *Smilaeaceae*, *Verbenaceae* [53,54]. Its activity was assessed for both its biological and pharmacological effects. The compound is a very powerful antioxidant [55]; it also has anti-inflammatory and analgesic [56], immunosuppressive [57], immunomodulating [58], anticancer [59], hepatoprotective [60] and antimicrobial [61] properties.

Inflammation, which is the process of faster aging of tissues, including skin tissue, caused by constant exposure to factors causing inflammation is a recently noticed and investigated problem. Counteracting the progressive effects of inflammation may consist in avoiding and reducing the frequency of exposure to external factors, causing inflammatory microstates, such as, for example, ultraviolet radiation, exhaust fumes, cigarette smoke and detergents, although it is not a simple task. Scientists place their hope in plants rich in certain specific glycosides, such as verbascoside, with anti-inflammatory and antioxidant effects. Such plants include: Plantain leaves with the amount of 3.5% of dry weight of verbascoside, verbena herb 3.7%, 12.9% of dry leaves in the fourth stage of *Sesamum indicum* L., and 27.7% of *Cymbaria daurica* L., which is one of the highest percentages reported in plants from nature. By comparison, the content of this compound and its derivatives in leaves, twigs, flowers and fruits of Oxytree are 9.47, 22.93, 2.31 and 17.91 mg/g DM, respectively.

In Oxytree, cistanoside F, which is also phenylpropanoid glycoside, was identified. The first time it was isolated and its structure was determined in the herbs of *Cistanche deserticola* Ma. was in 1985 by Kobayashi et al. [27] The concentration of this compound in *Sesamum indicum* L. was 0.26%, in lemon verbena 2%, and in Oxytree from 0.47% in twigs to 0.02% in flowers.

Cistanoside F shows vasorelaxant and antioxidative effects; it shows a strong free radical scavenging activity on 1,1-diphenyl-2-picrylhydrazyl (DPPH) radical and xanthine/xanthine oxidase (XOD) generated superoxide anion radical (O_2_^−^**^·^**). Since cistanoside F contains only one phenolic ring in the caffeoyl moiety, it has lower antioxidant activity that compounds with two aromatic rings. The highest content of this compound—4.28 mg/g DM—was found in twigs, 1.92 mg/g was found in fruit and 1.22 mg/g in leaves, and the least—0.54 mg/g DM—in Oxytree flowers.

Clerodendrin is used in the manufacture of a medicament for the treatment of degenerative changes in the retina. Clerodendrin showed a hypotensive activity in rats by measurement of the tail volume pressure in preliminary pharmacological tests. This compound belongs to the flavone glycosides with the structure of apigenin7-*O*-digiucuronide and for the first time has been isolated from the leaves of *Clerodendron trichotomum* Thunb. It is possible to obtain this compound from *Scutellaria barbata* and *Glechoma longituba* (Nakai) Kupr. A method for producing clerodendrin isolated from *Clerodendron trichotomum* Folium and a pharmaceutical composition containing the above compound for the prevention and treatment of gastritis and reflux esophagitis has been patented (PCT/KR2003/001037). Clerodendrin was identified in trace amounts, while other flavonoids—apigenin glucuronide, apigenin-7-*O*-glucuronopyranosyl(1→2)-glucuronopyranoside and luteolin-diglucuronide—were in total determined in small amounts only in leaves and flowers: 1.61 and 1.99 mg/g DM.

The third and last group of organic chemicals that belong to terpenes described in Oxytree are the iridoids. The representative of this group with the highest concentration is catalpol—simply monoterpene with a glucose molecule attached. It was primarily obtained from the root of *Rehmannia glutinosa* Libosch. (1–10% depending on the variety) and also was isolated from the *Cymbaria daurica* L. (11.4%), *Plantago lanceolate*, *Buddleia* species, Radix scrophulariae and *Lancea tibetica*. Pharmacological effects include analgesic, sedative, hepatoprotective, laxative, anti-inflammatory, antibacterial, anti-tumor and anti-apoptotic effects. In Oxytree, this compound is present in all examined morphological parts, i.e., in leaves, twigs, flowers and fruits, in concentrations 0.65, 9.96, 3.08 and 7.36 mg/g DM.

Another iridoid is 7-hydroxytomentoside, about which there is very little information in the literature. It was first isolated from *Paulownia tomentosa* and described in 1993 by Damtoft and Jensen [62]. It was then isolated from *Paulownia coreana* Uyeki leaves. The 7-hydroxytomentoside has anti-inflammatory effects on skin with irritations, atopy and acne. It is used to lighten the skin and reduce the signs of aging. From all the morphological parts of Oxytree, 7-hydroxytomentoside is found only in leaves at a rate of 6 mg/g DM.

Even if the concentrations of the described compounds in Oxytree are not spectacular, due to the fact that the number of leaves, twigs and even flowers on the trees is so high and the tree density in the plantation is 600 pcs/ha, they can be an excellent source of biologically active natural phytochemicals used in medicine and cosmetology.

So far, only a few studies focusing on biological activity of preparations derived from organs of *Paulownia* Clon in Vitro 112 have been carried out; therefore, the aim of this study was to investigate the effect of the extract and its four fractions (A–D) derived from *Paulownia* Clon in Vitro 112 leaves concerning the process of thrombus formation using the T-TAS technique in whole blood collected from healthy volunteers. The first time we have demonstrated that all tested preparations showed anticoagulant activity. The tested extract and four fractions (A–D) decreased AUC10 relative to control (blood without extract/fraction); however, these effects were not always statistically significant. Only three fractions (A, C and D) at the highest concentration—50 µg/mL—induced changes which were statistically significant. In addition, the fraction D (50 µg/mL), rich in triterpenoids, showed the strongest activity. Such a strong effect may be associated with the presence of triterpenoids, including maslinic, pomoleic, ursolic and 3-epiurolic acid. For example, triterpenoids and acylated triterpenoids isolated from sea buckthorn had a greater impact on the coagulation system than phenolic compounds [63]. Results of Babalola et al. [64] also demonstrated that naturally occurring triterpenoids—oleanolic, ursolic, betulinoic and maslinic acids—inhibit platelet aggregation. Three platelet agonists, i.e., thrombin, ADP or epinephrine, were used to assess the anti-aggregation activity of these triterpenes in a rat model. Inhibition of platelet aggregation stimulated by thrombin and ADP was most demonstrated by oleanolic acid (approximately 80%). In other studies, Kim et al. [65] observed that ursolic and oleanolic acids enhanced platelet aggregation induced by thrombin and ADP in a dose-dependent manner—the strongest effect was observed for the concentration of 50 μM (for both used acids). However, after the activation of platelets by collagen, the acids had no effect on their aggregation. Incubation of whole blood with thrombin, in the presence of tested acids, also showed proaggregation activity compared to the control sample. In addition, measurement of prothrombin time and the activated partial thromboplastin time showed that these compounds did not affect plasma coagulation. Results of Kim et al. [23] may indicate that blood platelets, not coagulation factors, are the main target for ursolic and olenolic acid. Moreover, our present results suggest that triterpenoids, including ursolic from used fraction D, may determine its anticoagulant potential. It is interesting that the concentration of this acid in tested fraction D is lower (<50 µg/mL) than its concentration (50 µM) used by Kim et al. [65]. In addition, the differences in the anti-coagulant potential of tested extract and fractions (observed in thrombus formation) might be correlated with their different chemical profile. Our present results indicate that fraction D has stronger anti-coagulant properties than extract (rich in verbascoside) in whole blood. On the other hand, when we used washed blood platelets, this extract had stronger anti-platelet activity than various fractions (A–D) [47]. Campo et al. [66] also observed that verabascocide inhibits platelet aggregation stimulated by ADP and arachidonic acid in vitro.

On the basis of the conducted research, it has been shown that the chemical compounds produced by *Paulownia* Clon in Vitro 112 leaves, especially presented in fraction D, are promising candidates for natural preparations with anticoagulant potential. However, in order to clarify the exact mechanism of action of the active substances present in this plant, studies closer to physiological conditions, i.e., in vivo studies, should be performed, which will also allow to determine the effects of their long-term effects. Moreover, its high calorific value makes Oxytree a good raw material to be used as a renewable carbon source. This species is ideal for establishing commercial forests.

## Figures and Tables

**Figure 1 molecules-27-00980-f001:**
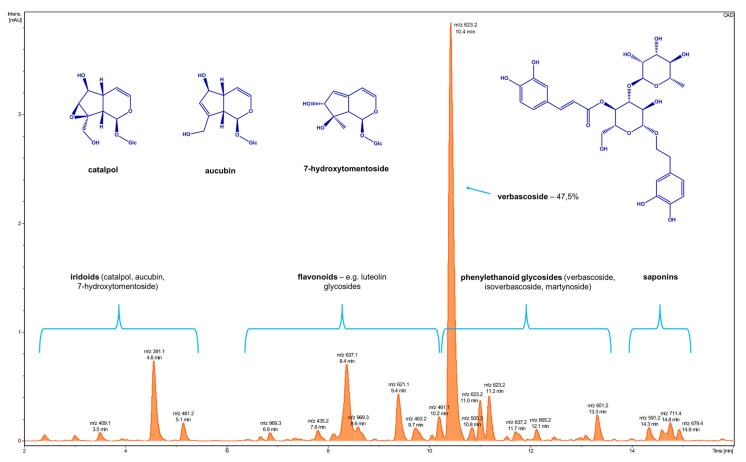
The UHPLC-CAD profile of a Paulownia Clon In Vitro 112^®^ extract of leaves from Łęka.

**Figure 2 molecules-27-00980-f002:**
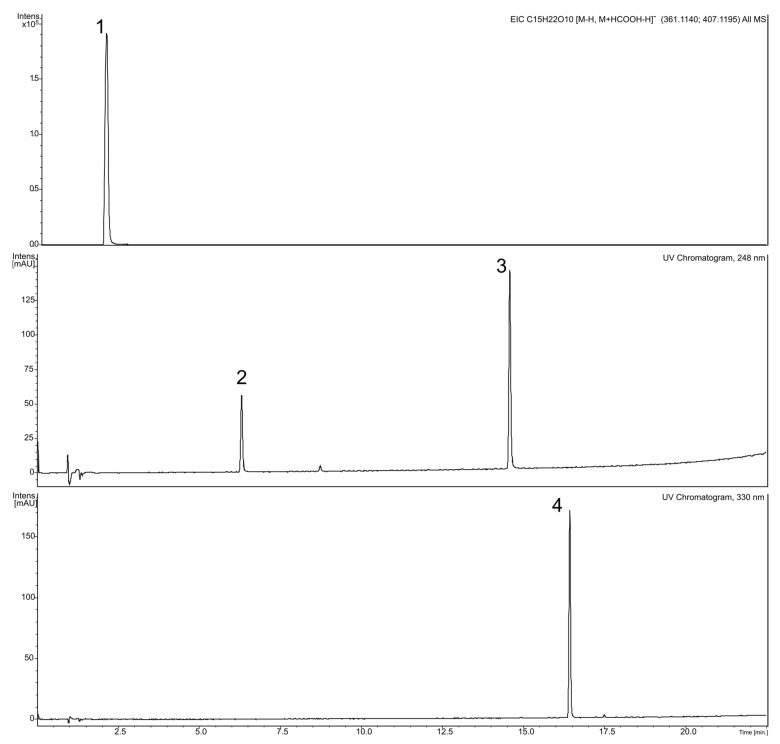
Chromatograms of isolated standards: 1—catalpol; 2—7-hydroxytomentoside; 3—apigenin-7-*O*-glucuronopyranosyl-(1→2)-glucuronopyranoside); 4—verbascoside.

**Figure 3 molecules-27-00980-f003:**
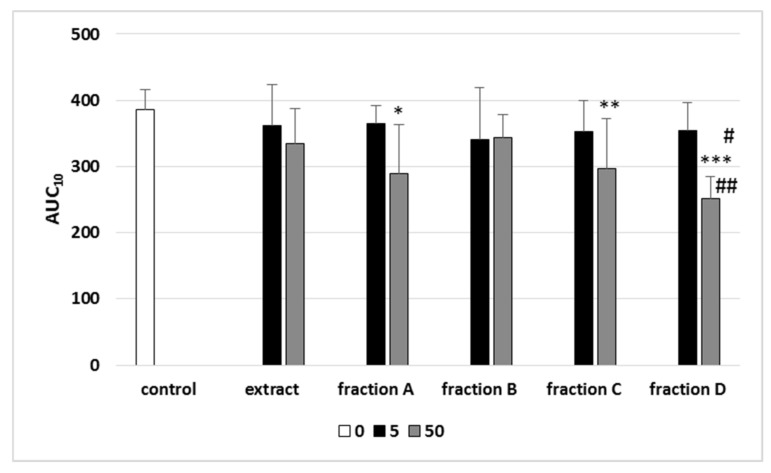
Effects of extract and four fractions (A–D) from Paulownia Clon In Vitro 112^®^ leaves (5 and 50 µg/mL; 30 min) on the T-TAS using the PL-chip in whole blood samples. Whole blood samples were analyzed by the T-TAS at the shear rates of 1000 s^−1^ on the PL-chips. The area under the curve (AUC_10_) in PL is shown. Data represent the means SD of six healthy volunteers. * *p* < 0.05 vs. control; ** *p* < 0.01; *** *p* < 0.001; # *p* < 0.01 (fraction D vs. fraction B); ## *p* < 0 05 (fraction D vs. extract).

**Figure 4 molecules-27-00980-f004:**
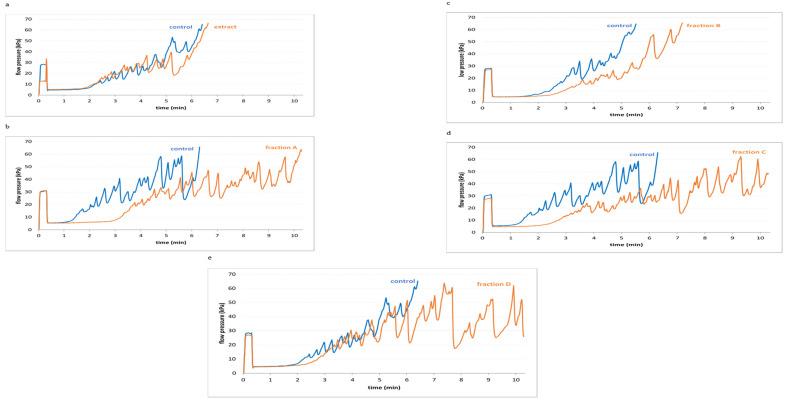
Selected diagrams for T-TAS analysis for extract (**a**) and four fractions (A–D) (**b**–**e**) from Paulownia Clon In Vitro 112^®^ leaves (50 µg/mL; 30 min).

**Table 1 molecules-27-00980-t001:** Secondary metabolites found in the leaves, bark, flowers and fruits of *Paulownia tomentosa var. tomentosa*.

Compound	The Morphological Part of the Plant				Reference
	Leaves	Bark	Fruits	Flowers	
matteucinol (syn. 4-*O*-methylfarrerol)	+				[1]
ursolic acid	+				
homoeriodictyol (syn. hesperetin)	+				[2]
3-epiursolic acid	+				
pomolic acid	+				
corosolic acid	+				
maslinic acid	+				
β-sitosterol	+				
Daucosterol	+				
Apigenin	+				[3]
Luteolin	+				[4]
Quercetin	+				
(+)-catechin	+				
(−)-epicatechin	+				
Naringenin	+				
Taxifolin	+				
7,3′-dimethylquercetin (syn. rhamnazin)	+				[5]
7,3′,4′-trimethylquercetin	+				
7,3′,4′-trimethylmyricetin	+				
7,3′,4′,5′-tetramethylmyricetin	+		+		
Diplacol	+			+	[6]
3′-*O*-methyldiplacone	+			+	
3′-*O*-methyl-5′-hydroxydiplacone (syn. 6-geranyl-4′,5,5′,7-tetrahy-droxy-3′-methoxyflavanone	+				
acteoside (syn. verbascoside)	+				[7]
isoacteoside (syn. isoverbascoside)	+				
7-β-hydroxyharpagide	+				[8]
Paulovnioside	+				
Catalpol	+				
Aucubin	+				
Tomentoside	+				
7-hydroxytomentoside	+				
*p*-hydroxybenzoic acid	+				[9]
vanillic acid	+				
gallic acid	+	+			
cinnamic acid	+	+			
*p*-coumaric acid	+				
caffeic acid	+				
Quercetin		+			[10]
Naringenin		+			
7-caffeoyl-acacetin (syn. 7-caffeoyl-4′-methoxyapigenin)		+			
isoacteoside (syn. isoverbascoside)		+			
isocampneoside II		+			
cistanoside F		+			
ilicifolioside A		+			[11]
campneoside II (syn. β-hydroxyacteoside)		+			
isoilicifolioside A		+			
isocampneoside I		+			[12]
coniferin (syn. abietin)		+			[13]
syringin (syn. eleutherosid B)		+			
acteoside (syn. verbascoside)		+			
β-oxoacteoside (syn. tomentoside A)		+			[14]
Martynoside		+			
campneoside I		+			
Catalpol		+			[15]
Dihydrotricin			+		[16]
6-isopentenyl-3′-*O*-methyltaxifolin			+		
3′-*O*-methyl-5′-hydroxydiplacone (syn. 6-geranyl-4′,5,5′,7-tetra-hydroxy-3′-methoxyflavanone)			+		
3′-*O*-methyl-5′-methoxydiplacol, schizolaenone C			+		
6-geranyl-3′,5,5′,7-tetrahydroxy-4′-methoxyflavanone			+		
tomentodiplacone B			+		
Tomentodiplacone			+		
Tomentodiplacol			+		
acteoside (syn. verbascoside)			+		
isoacteoside (syn. isoverbascoside)			+		
3,4′,5,5′,7-pentahydroxy-3′-methoxy-6-(3-methyl-2-butenyl)flavanone			+		[17]
Diplacol			+		
3′-*O*-methyldiplacone, 3′-*O*-methyldiplacol (syn. diplacol 3′-*O*-methylether)			+		
3′-*O*-methyl-5′-*O*-methyldipla-cone (syn. 6-geranyl-4′,5,7-tri-hydroxy-3′,5′-dimethoxyflavanone			+		
6-geranyl-5,7-dihydroxy-3′,4′-dimethoxyflavanone			+		
3,3′,4′,5,7-pentahydroxy-6-[7-hydroxy-3,7-dimethyl-2(*E*)octenyl]flavanone			+		
Prokinawan			+		
4′,5,5′,7-tetrahydroxy-6-[6-hydroxy-3,7-dimethyl-2(*E*),7-octadienyl]-3′-methoxyflavanone			+		
3,3′,4′,5,7-pentahydroxy-6-[6-hydroxy-3,7-dimethyl-2(*E*),7-octadienyl]flavanone			+		
6-geranyl-3′,5,7-trihydroxy-4′-methoxyflavanone (syn. 4′-*O*-methyldiplacone)			+		[18]
6-geranyl-3,3′,5,7-tetrahydroxy-4′-methoxyflavanone (syn. 4′-*O*-methyldiplacol)			+		
6-geranyl-3,3′,5,5′,7-pentha-hydroxy-4′-methoxyflavanone			+		
tomentin A, B, C, D, E			+		
tanariflavanone D			+		[19]
Tomentomimulol			+		
mimulone B			+		
mimulone C, D, E,			+		[20]
tomentodiplacone C, D, E, F, G, H, I			+		
5,7-dihydroxy-6-geranylchromone			+		[16]
Apigenin				+	[21]
Mimulone (syn. 6-geranylnaringenin)				+	
5,4′-dihydroxy-7,3′-dimethoxyflavanone,				+	
diplacone (syn. propolin C)				+	
5-hydroxy-7,3′,4′-trimethoxyflavanone				+	[22,23]
isoatriplicolide tiglate				+	
3′-*O*-methyldiplacol (syn. diplacol 3′-*O*-methylether)				+	[6]
Prokinawan				+	
*p*-ethoxybenzaldehyde				+	[24]

**Table 2 molecules-27-00980-t002:** Concentration of compounds and class of compounds in leaves, twigs, flowers and fruits of Paulownia Clon in Vitro.

Compound/Class of Compounds	Concentration [mg/g DM]			
	Leaves	Twigs	Flowers	Fruits
catalpol	0.65 ± 0.06	9.96 ± 1.23	3.08 ± 0.20	7.36 ± 0.02
7-hydroxytomentoside	6.08 ± 0.21	+	+	+
cistanoside F	0.61 ± 0.01	2.14 ± 0.17	0.26 ± 0.02	0.95 ± 0.04
cistanoside F isomer	0.61 ± 0.02	2.14 ± 0.15	0.28 ± 0.02	0.97 ± 0.06
luteolin-diglucuronide	0.79 ± 0.07	+	+	+
hydroxyverbascoside I	2.16 ± 0.25	+	+	4.47 ± 0.21
hydroxyverbascoside II	2.88 ± 0.23	6.87 ± 0.12	0.65 ± 0.08	4.78 ± 0.23
apigenin-7-*O*-glucuronopyranosyl (1→2)-glucuronopyranoside	0.82 ± 0.07	+	1.02 ± 0.03	+
oxoverbascoside	0.90 ± 0.11	+	1.23 ± 0.05	0.11 ± 0.01
methoxyverbascoside	2.62 ± 0.30	4.96 ± 0.23	0.44 ± 0.09	6.14 ± 0.46
dimethylverbascoside	0.90 ± 0.03	+	+	+
verbascoside	+	11.11 ± 1.55	0.97 ± 0.17	2.41 ± 0.37
apigenin glucuronide	+	+	+	+
Iridoids	6.73	9.96	3.08	7.36
Phenylethanoid glycosides	10.69	27.21	2.85	19.83
Flavonoids	1.61	+	1.99	+

+ trace amount.

## Data Availability

Data supporting reported results can be found: https://www.iung.pl/o-instytucie/struktura/dzialy-wspomagania/karta-dwb/ (accessed on 14 December 2021).

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
