# Peer review of "Qualitative and Quantitative Analysis of Secondary Metabolites in Morphological Parts of Paulownia Clon In Vitro 112® and Their Anticoagulant Properties in Whole Human Blood"

_molecules, 2022, doi:10.3390/molecules27030980_

Round 1
Reviewer 1 Report
The manuscript submitted by Stochmal and co-workers is a compilation of interesting data of biological activity of several secondary metabolites, but far from a publishable work.
The introduction begins with a detailed compilation of secondary metabolites identified in different parts of Paulownia tomentosa var. tomentosa. However, it suddenly starts the list of secondary metabolite groups and their biological activities.
What do the authors looking with this?
All the compounds identified in Paulownia tomentosa var. tomentosa exhibit the biological activity described for secondary metabolites groups?
For example, all the flavonoids identified exhibit anti-cancer activity. If that happens, please describe it precisely; if not, what is the meaning of the introduction?
Why does the extraction of the compounds and the qualitative and quantitative analysis of the morphological parts come from different trees of different ages?
The authors should include the composition of the extract and fractions A, B, C, and D used in the T-TAS assay as shown for those extracts used for the development of analytical methods.
Besides, they need to compare both systems, considering that they come from different samples of different ages.
The authors mention that the most widespread Oxytree compound is verbascoside. Why not use twigs or fruits extracts instead of leave extracts?
Figure 3. The area under the 293 curve (AUC10) in PL is shown as closed circles. This part of the plot is missing.
There is no discussion about the results shown in Figures 3 and 4. Each plot from figure 4, What does it mean?
Only Extract C and D at a 50 ug/ml concentration showed a statistical difference with the control on the T-TAS assay. The discussion section mentioned that ursolic and oleanolic acid-enhanced platelet aggregation induced by thrombin in a dose-dependent manner, showing the strongest effect at 50 uM. With these results, the authors suggest that blood platelets, not coagulation factors, are the main target for ursolic and oleanolic acid.
Does the ursolic acid concentration in the fractions C and D are close to 50 uM?
Are the rest of the compounds identified in those fractions innocuous for thrombus formation?
The results do not support the conclusions provided by the authors. Therefore I recommend not publishing this manuscript in Molecules.
Author Response
The manuscript submitted by Stochmal and co-workers is a compilation of interesting data of biological activity of several secondary metabolites, but far from a publishable work.
The introduction begins with a detailed compilation of secondary metabolites identified in different parts of Paulownia tomentosa var. tomentosa. However, it suddenly starts the list of secondary metabolite groups and their biological activities.
What do the authors looking with this?
All the compounds identified in Paulownia tomentosa var. tomentosa exhibit the biological activity described for secondary metabolites groups?
For example, all the flavonoids identified exhibit anti-cancer activity. If that happens, please describe it precisely; if not, what is the meaning of the introduction?
Why does the extraction of the compounds and the qualitative and quantitative analysis of the morphological parts come from different trees of different ages?
Response: First, we started isolation works using leaves collected from one-year-old trees on a plantation in the Lublin region. Then we wanted to compare the content of compounds in different morphological parts of Oxytrees, but they only bloom after three years of cultivation, in spring at the beginning of the fourth year. Therefore, material was collected for research from an older plantation in Wielkopolska.
The authors should include the composition of the extract and fractions A, B, C, and D used in the T-TAS assay as shown for those extracts used for the development of analytical methods.
Besides, they need to compare both systems, considering that they come from different samples of different ages.
The authors mention that the most widespread Oxytree compound is verbascoside. Why not use twigs or fruits extracts instead of leave extracts?
Response: The leaf samples were only used for compound isolation. It was not our intention to compare changes in compounds levels under biotic or abiotic stress.
It is not possible to compare samples from both plantations, because the temperature in the Lublin region in winter is much lower and although the trees have reached flowering maturity, flower buds freeze.
Figure 3. The area under the 293 curve (AUC10) in PL is shown as closed circles. This part of the plot is missing.
There is no discussion about the results shown in Figures 3 and 4. Each plot from figure 4, What does it mean?
Response: We have added more information about it: “The first time, we have demonstrated that all tested preparations showed anticoagulant activity. Tested extract and four fractions (A-D) decreased AUC10 relative to control (blood without (extract/fraction), however these effects were not always statistically significant. Only three fractions (A, C and D) at the highest concentration – 50 µg/mL induced changes, which were statistically significant. In addition, the fraction D (50 µg/mL) rich in triterpenoids, showed the strongest activity.” (chapter of discussion).
We have also added other information: “Data represent (on Fig. 3) mean ± SD of six healthy volunteers (each experiment performed in triplicate). Figures 4 demonstrates selected diagram for tested extract and tested fractions (A-D). We have calculated AUC10 for each diagram.” (chapter of results).
In addition, we have added more information about T-TAS method: “The results were obtained as AUC10 (Area Under the Curve) using a PL chip. The AUC10 parameter that is an area under the pressure curve from the start of the test to a time of 10 min was studied. This parameter determine the growth, intensity, and stability of thrombus formation. A more detailed description of this method can be found in Hosokawa et al. (2011).”
Figure 3. The area under the curve (AUC10) in PL are shown as value of the bars.
Only Extract C and D at a 50 ug/ml concentration showed a statistical difference with the control on the T-TAS assay. The discussion section mentioned that ursolic and oleanolic acid-enhanced platelet aggregation induced by thrombin in a dose-dependent manner, showing the strongest effect at 50 uM. With these results, the authors suggest that blood platelets, not coagulation factors, are the main target for ursolic and oleanolic acid.
Does the ursolic acid concentration in the fractions C and D are close to 50 uM?
Are the rest of the compounds identified in those fractions innocuous for thrombus formation?
The results do not support the conclusions provided by the authors. Therefore I recommend not publishing this manuscript in Molecules.
Response: We have added more information about ursolic acid: “Results of Kim et al. (2014) may indicate that blood platelets, not coagulation factors, are the main target for ursolic and olenolic acid. Moreover, our present results suggest that triterpenoids, including ursolic from used fraction D may decide about its anticoagulant potential. It is an interesting that the concentration of this acid in tested fraction D is lower (<50 µg/mL) then its concentration (50 µM) used by Kim et al. (2014). In addition, the differences in the anti-coagulant potential of tested extract and fractions (observed in thrombus formation) might be correlated with their different chemical profile.” (chapter of discussion).
Reviewer 2 Report
Manuscript ID: molecules-1532305
Title: Qualitative and quantitative analysis of secondary metabolites in morphological parts of Paulownia Clon in Vitro 112® and their anticoagulant properties in whole human blood
Major revisions:
- Pag 2, lines 51-93: I suggest presenting these data in a table and, if possible, grouping them in the classes of the respective compounds, and their respective references.
- The literature already describe that some compound of the wild plant features anticoaguant/antiplatelet activity?
- Pag 3, Line 123: the data were published? (add reference of the study).
- Pag 4, Line 153 (2.3. Extraction and isolation of compounds): what the proportion of plant material (w/v)? - Line 155.
- Explain the yields of the metanol total extract (41.7%) and in follow stage was 44.1% (Line 165), was pooled the two fractions 1% and 80% metanol?
- Pag 4, Lines 173-178: Where is the data for the components found in fractions B, C and D presented?
- In figure 2: the compounds 2 iridoid glycosides – catalpol [5] (as a mixture) and 7-hydroxytomentoside 261 [6] not were shown (Pag 6, Lines 261-262). In supplementary material, point in each graphical what sample/fraction. In Figure 2 was demonstrate only the chromatograms of isolated standards? And of the samples studied/analyzed?
- Figure 1: indicate in legend the material type (extract/fraction).
- Line 238-239: replace µl by µL (standardize in the text).
- In methods: °C or ° C - standardize in the text.
- What is the difference between the compounds found in Paulownia Clon Leka and Września leaves? Include all the data of the leaves composition in Table 2.
- Standardize in the text Fig. or Figure, according to the rules.
- Pag 7, Line 278: suppress the tested word that is repeated.
- Figure 3: Why the fraction A at the concentration of 50 was not significant, since the very similar fraction C was significant, compared to the control? see the description of the results. Include a positive control.
- Pag 8, Lines 240, 293-294: explain why AUC10 (in methods) and the sentence in the legend of Figure 8 “the area under the curve (AUC 10) in PL are shown as closed circles”?
- Explain/discuss why the extract (50 µg/mL) did not show a significant anticoagulant effect in the thrombogenic test, since the fractions (50 µg/mL) obtained from the extract showed an anticoagulant effect, especially the fraction D.
- Pag 10, Line 309-314: I suggest suppress.
- Pag 12, Line 308: “fraction D rich in triterpenoids” - was analyzed in this study? or add reference.
- Is there any data in the literature correlating the effect on platelet coagulation/aggregation and the verbascoside major compound?
- Suggestion: Include conclusion.
- Pag 4, Line 179 (2.4. NMR Spectroscopy): Add the samples that were analyzed.
- Pag 4, Line 190: all extracts of the differents parts were obtained by same method of extraction? Add this information.
- Why the leaves of the “Leka” plant were collected at one year of age and the parts (leaves, twigs, flowers and fruits) used in the extractions for quantitative and qualitative studies were collected at five years of age (Września-Wielkopolska-)?
Author Response
Major revisions:
- Pag 2, lines 51-93: I suggest presenting these data in a table and, if possible, grouping them in the classes of the respective compounds, and their respective references.
Response: As suggested by the Reviewer, the table was prepared and included in the publication. We have added new Table 1.
Table 1. Secondary metabolites found in the leaves, bark, flowers and fruits of Paulownia tomentosa var. tomentosa.
Compound |
The morphological part of the plant |
Reference |
||||
Leaves |
Bark |
Fruits |
Flowers |
|
|
|
matteucinol (syn. 4-O-methylfarrerol) |
+ |
|
|
|
Zhu et al., 1986 |
|
ursolic acid |
+ |
|
|
|
|
|
homoeriodictyol (syn. hesperetin) |
+ |
|
|
|
Zhang and Li, 2011 |
|
3-epiursolic acid |
+ |
|
|
|
|
|
pomolic acid |
+ |
|
|
|
|
|
corosolic acid |
+ |
|
|
|
|
|
maslinic acid |
+ |
|
|
|
|
|
β-sitosterol |
+ |
|
|
|
|
|
daucosterol |
+ |
|
|
|
|
|
apigenin |
+ |
|
|
|
Zhao et al., 2012 |
|
luteolin |
+ |
|
|
|
Si et al., 2008a |
|
quercetin |
+ |
|
|
|
|
|
(+)-catechin |
+ |
|
|
|
|
|
(-)-epicatechin |
+ |
|
|
|
|
|
naringenin |
+ |
|
|
|
|
|
taxifolin |
+ |
|
|
|
|
|
7,3’-dimethylquercetin (syn. rhamnazin) |
+ |
|
|
|
Wollenweber et al., 2008 |
|
7,3’,4’-trimethylquercetin |
+ |
|
|
|
|
|
7,3’,4’-trimethylmyricetin |
+ |
|
|
|
|
|
7,3’,4’,5’-tetramethylmyricetin |
+ |
|
+ |
|
|
|
diplacol |
+ |
|
|
+ |
Kobayashi et al., 2008 |
|
3’-O-methyldiplacone |
+ |
|
|
+ |
|
|
3’-O-methyl-5’-hydroxydiplacone (syn. 6-geranyl-4’,5,5’,7-tetrahy- droxy-3’-methoxyflavanone |
+ |
|
|
|
|
|
acteoside (syn. verbascoside) |
+ |
|
|
|
Schilling et al., 1982 |
|
isoacteoside (syn. isoverbascoside) |
+ |
|
|
|
|
|
7-β-hydroxyharpagide |
+ |
|
|
|
Adriani et al., 1981 |
|
paulovnioside |
+ |
|
|
|
|
|
catalpol |
+ |
|
|
|
|
|
aucubin |
+ |
|
|
|
|
|
tomentoside |
+ |
|
|
|
|
|
7-hydroxytomentoside |
+ |
|
|
|
|
|
p-hydroxybenzoic acid |
+ |
|
|
|
Ota et al., 1993 |
|
vanillic acid |
+ |
|
|
|
|
|
gallic acid |
+ |
+ |
|
|
|
|
cinnamic acid |
+ |
+ |
|
|
|
|
p-coumaric acid |
+ |
|
|
|
|
|
caffeic acid |
+ |
|
|
|
|
|
quercetin |
|
+ |
|
|
Si et al., 2011b |
|
naringenin |
|
+ |
|
|
|
|
7-caffeoyl-acacetin (syn. 7-caffeoyl-4’-methoxyapigenin) |
|
+ |
|
|
|
|
isoacteoside (syn. isoverbascoside) |
|
+ |
|
|
|
|
isocampneoside II |
|
+ |
|
|
|
|
cistanoside F |
|
+ |
|
|
|
|
ilicifolioside A |
|
+ |
|
|
Si et al., 2008b |
|
campneoside II (syn. β-hydroxyacteoside) |
|
+ |
|
|
|
|
isoilicifolioside A |
|
+ |
|
|
|
|
isocampneoside I |
|
+ |
|
|
Si et al., 2008d |
|
coniferin (syn. abietin) |
|
+ |
|
|
Sticher and Lahloub, 1982 |
|
syringin (syn. eleutherosid B) |
|
+ |
|
|
|
|
acteoside (syn. verbascoside) |
|
+ |
|
|
|
|
β-oxoacteoside (syn. tomentoside A) |
|
+ |
|
|
Kang et al., 1994 |
|
martynoside |
|
+ |
|
|
|
|
campneoside I |
|
+ |
|
|
|
|
catalpol |
|
+ |
|
|
Plouvier, 1971 |
|
dihydrotricin |
|
|
+ |
|
Šmejkal et al., 2007a |
|
6-isopentenyl-3’-O-methyltaxifolin |
|
|
+ |
|
|
|
3’-O-methyl-5’-hydroxydiplacone (syn. 6-geranyl-4’,5,5’,7-tetra- hydroxy-3’-methoxyflavanone) |
|
|
+ |
|
|
|
3’-O-methyl-5’-methoxydiplacol, schizolaenone C |
|
|
+ |
|
|
|
6-geranyl-3’,5,5’,7-tetrahydroxy-4’-methoxyflavanone |
|
|
+ |
|
|
|
tomentodiplacone B |
|
|
+ |
|
|
|
tomentodiplacone |
|
|
+ |
|
|
|
tomentodiplacol |
|
|
+ |
|
|
|
acteoside (syn. verbascoside) |
|
|
+ |
|
|
|
isoacteoside (syn. isoverbascoside) |
|
|
+ |
|
|
|
3,4’,5,5’,7-pentahydroxy-3’ -methoxy-6-(3-methyl-2-butenyl)flavanone |
|
|
+ |
|
Asai et al., 2008 |
|
diplacol |
|
|
+ |
|
|
|
3’-O-methyldiplacone, 3’-O-methyldiplacol (syn. diplacol 3’-O-methylether) |
|
|
+ |
|
|
|
3’-O-methyl-5’-O-methyldipla- cone (syn. 6-geranyl-4’,5,7-tri- hydroxy-3’,5’-dimethoxyflavanone |
|
|
+ |
|
|
|
6-geranyl-5,7-dihydroxy-3’,4’-dimethoxyflavanone |
|
|
+ |
|
|
|
3,3’,4’,5,7-pentahydroxy-6-[7-hydroxy-3,7-dimethyl-2(E)octenyl]flavanone |
|
|
+ |
|
|
|
prokinawan |
|
|
+ |
|
|
|
4’,5,5’,7-tetrahydroxy-6-[6-hydroxy-3,7-dimethyl-2(E),7-octadienyl]-3’-methoxyflavanone |
|
|
+ |
|
|
|
3,3’,4’,5,7-pentahydroxy-6-[6-hydroxy-3,7-dimethyl-2(E),7-octadienyl]flavanone |
|
|
+ |
|
|
|
6-geranyl-3’,5,7-trihydroxy-4’-methoxyflavanone (syn. 4’-O-methyldiplacone) |
|
|
+ |
|
Cho et al., 2012, 2013 |
|
6-geranyl-3,3’,5,7-tetrahydroxy-4’-methoxyflavanone (syn. 4’-O-methyldiplacol) |
|
|
+ |
|
|
|
6-geranyl-3,3’,5,5’,7-pentha- hydroxy-4’-methoxyflavanone |
|
|
+ |
|
|
|
tomentin A, B, C, D, E |
|
|
+ |
|
|
|
tanariflavanone D |
|
|
+ |
|
Schneiderova´et al., 2013 |
|
tomentomimulol |
|
|
+ |
|
|
|
mimulone B |
|
|
+ |
|
|
|
mimulone C, D, E, |
|
|
+ |
|
Navrátilova´ et al., 2013 |
|
tomentodiplacone C, D, E, F, G, H, I |
|
|
+ |
|
|
|
5,7-dihydroxy-6-geranylchromone |
|
|
+ |
|
Šmejkal et al., 2008a |
|
apigenin |
|
|
|
+ |
Jiang et al., 2004 |
|
mimulone (syn. 6-geranylnaringenin) |
|
|
|
+ |
|
|
5,4’-dihydroxy-7,3’-dimethoxyflavanone, |
|
|
|
+ |
|
|
diplacone (syn. propolin C) |
|
|
|
+ |
|
|
5-hydroxy-7,3’,4’-trimethoxyflavanone |
|
|
|
+ |
Kim et al., 2010a, b |
|
isoatriplicolide tiglate |
|
|
|
+ |
|
|
3’-O-methyldiplacol (syn. diplacol 3’-O-methylether) |
|
|
|
+ |
Kobayashi et al., 2008 |
|
prokinawan |
|
|
|
+ |
|
|
p-ethoxybenzaldehyde |
|
|
|
+ |
Yuan et al., 2009 |
|
- The literature already describe that some compound of the wild plant features anticoaguant/antiplatelet activity?
Response: Two papers; Adach, W., Żuchowski, J., Moniuszko-Szajwaj, B., Szumacher=Strabel, M., Stochmal, A., Olas, B., Cieślak, A. (2020). Comparative phytochemical, antioxidant, and hemostatic studies of extract and four fractions from Paulownia Clone in Vitro 112 leaves in human plasma. Molecules 25, 1-14.
Adach, W., Żuchowski, J., Moniuszko-Szajwaj, B., Szumacher-Strabel, M., Stochmal, A., Olas, B., Cieślak, A. (2020). In vitro antiplatelet activity of extract and its fractions of Paulownia Clone in Vitro 112 leaves. Biomed. Pharmacother. 137, 1-10.
described the effect of preparations from Pulownia leaves on hemostasis using human plasma and washed blood platelets. This present manuscript shows the effect of these preparations on hemostasis using human whole blood.
- Pag 3, Line 123: the data were published? (add reference of the study).
Response: References to both publications are attached (one of them is in Molecules).
- Pag 4, Line 153 (2.3. Extraction and isolation of compounds): what the proportion of plant material (w/v)? - Line 155.
Response: The 100 g of plant material was extracted with 1000 mL, three times.
- Explain the yields of the metanol total extract (41.7%) and in follow stage was 44.1% (Line 165), was pooled the two fractions 1% and 80% metanol?
Response: Fraction 1% methanol was rejected as it contained sugars and not secondary metabolites.
- Pag 4, Lines 173-178: Where is the data for the components found in fractions B, C and D presented?
Response: This data are presented in: Adach, W., Żuchowski, J., Moniuszko-Szajwaj, B., Szumacher=Strabel, M., Stochmal, A., Olas, B., Cieślak, A. (2020). Comparative phytochemical, antioxidant, and hemostatic studies of extract and four fractions from Paulownia Clone in Vitro 112 leaves in human plasma. Molecules 25, 1-14
- In figure 2: the compounds 2 iridoid glycosides – catalpol [5] (as a mixture) and 7-hydroxytomentoside 261 [6] not were shown (Pag 6, Lines 261-262). In supplementary material, point in each graphical what sample/fraction. In Figure 2 was demonstrate only the chromatograms of isolated standards? And of the samples studied/analyzed?
Response: In Figure 2, compounds-standards are marked with different numbers than in the text: Figure 2. Chromatograms of isolated standards: 1 - catalpol; 2 - 7-hydroxytomentoside; 3 -apigenin-7-O-glucuronopyranosyl-(1→2)-glucuronopyranoside); 4 – verbascoside.
- Figure 1: indicate in legend the material type (extract/fraction).
Response: It was an extract and this information was included in the legend of figure 1.
- Line 238-239: replace µl by µL (standardize in the text).
Response: We have corrected.
- In methods: °C or ° C - standardize in the text.
Response: We have corrected.
- What is the difference between the compounds found in Paulownia Clon Leka and Września leaves? Include all the data of the leaves composition in Table 2.
Response: The content of compounds in the leaves from Łęka is irrelevant for the publication, because they are collected from trees that are much younger (in the first year of cultivation), and this problem was not the aim of our work. Oxytrees bud in their third year of growth and bloom in their fourth year in spring. The unfavorable winters in the Lublin region are too frosty and the flowers of my trees freeze.
- Standardize in the text Fig. or Figure, according to the rules.
Response: We have corrected. Now, it is “Fig.”
- Pag 7, Line 278: suppress the tested word that is repeated.
Response: The repeated word tested was suppressed and now the text reads: Tested extract and fractions (A-D)…
- Figure 3: Why the fraction A at the concentration of 50 was not significant, since the very similar fraction C was significant, compared to the control? see the description of the results. Include a positive control.
Response: We have corrected Fig. 3. Of course, the effect of fraction A (50 ug/mL) on AUC10 was statistically significant.
We used only one control (blood without extract/fraction).
- Pag 8, Lines 240, 293-294: explain why AUC10 (in methods) and the sentence in the legend of Figure 8 “the area under the curve (AUC 10) in PL are shown as closed circles”?
Response: We have added more information about T-TAS method: “The results were obtained as AUC10 (Area Under the Curve) using a PL chip. The AUC10 parameter that is an area under the pressure curve from the start of the test to a time of 10 min was studied. This parameter determine the growth, intensity, and stability of thrombus formation. A more detailed description of this method can be found in Hosokawa et al. (2011).”
Figure 3. The area under the curve (AUC10) in PL are shown as value of the bars.
- Explain/discuss why the extract (50 µg/mL) did not show a significant anticoagulant effect in the thrombogenic test, since the fractions (50 µg/mL) obtained from the extract showed an anticoagulant effect, especially the fraction D.
Response: We have added more information about it: “Results of Kim et al. (2014) may indicate that blood platelets, not coagulation factors, are the main target for ursolic and olenolic acid. Moreover, our present results suggest that triterpenoids, including ursolic from used fraction D decide about its anticoagulant potential. It is an interesting that the concentration of this acid in tested fraction D is lower (<50 µg/mL) then its concentration (50 µM) used by Kim et al. (2014). In addition, the differences in the anti-coagulant potential of tested extract and fractions (observed in thrombus formation) might be correlated with their different chemical profile. Our present results indicate that fraction D has stronger anti-coagulant properties then extract (rich in verbascoside) in whole blood. On the other hand, when we used washed blood platelets, this extract had stronger anti-platelet activity then various fractions (A-D) (Adach et al., 2021). Campo et al. al. (2012) also observed that verabascocide inhibits platelet aggregation stimulated by ADP and arachidonic acid, in vitro.” (chapter of discussion).
- Pag 10, Line 309-314: I suggest suppress.
Response: As suggested by the reviewer, the text fragment has been removed:
Verbascoside has numerous uses in cosmetology:
- helps to detangle tangled hair and gently cares for curly hair without irritating the hair,
- absorbs quickly into the skin and moisturizes it; cares for nails and cuticles,
- acts prophylactically against stretch marks, improves skin elasticity,
- provides remedy for swollen eyes, removes dark circles under the eyes.
- Pag 12, Line 308: “fraction D rich in triterpenoids” - was analyzed in this study? or add reference.
Response: Composition of fraction D was analyzed earlier and reference is added: Adach, W., Żuchowski, J., Moniuszko-Szajwaj, B., Szumacher=Strabel, M., Stochmal, A., Olas, B., Cieślak, A. (2020). Comparative phytochemical, antioxidant, and hemostatic studies of extract and four fractions from Paulownia Clone in Vitro 112 leaves in human plasma. Molecules 25, 1-14
- Is there any data in the literature correlating the effect on platelet coagulation/aggregation and the verbascoside major compound?
Response: We have added new publication about it (Campo et al., 2012).
- Suggestion: Include conclusion.
Response: We have added new information: “On the basis of the conducted research, it has been shown that the chemical compounds produced by Paulownia Clon in Vitro 112 leaves, especially presented in fraction D are promising candidates for natural preparations with anticoagulant potential.”
- Pag 4, Line 179 (2.4. NMR Spectroscopy): Add the samples that were analyzed.
Response: Information that these were isolated compounds was added.
- Pag 4, Line 190: all extracts of the differents parts were obtained by same method of extraction? Add this information.
Response: This information was added: Extracts and fractions of Paulownia leaves, flowers, fruit and twigs were prepared by the same method described above (2.3) and next were analyzed by…
- Why the leaves of the “Leka” plant were collected at one year of age and the parts (leaves, twigs, flowers and fruits) used in the extractions for quantitative and qualitative studies were collected at five years of age (Września-Wielkopolska-)?
Response: First, we started isolation works using leaves collected from one-year-old trees on a plantation in the Lublin region. Then we wanted to compare the content of compounds in different morphological parts of Oxytrees, but they only bloom after three years of cultivation, in spring at the beginning of the fourth year. Therefore, material was collected for research from an older plantation in Wielkopolska.
Round 2
Reviewer 2 Report
Minor revisions:
1. In results there are two tables 2.
2. Pg 14, Line 339: remove extra parenthesis.
3. I suggest removing the rows from inside the tables to be cleaner.
4. I suggest including the data of the calibration curves inside each figure, respectively, and deleting the table (Calibration curves of standards).
Author Response
Minor revisions:
- In results there are two tables 2
Response: As suggested by the Reviewer, one of the tables 2 has been removed. The standard curve equations can be found in Section 2.5.
Pg 14, Line 339: remove extra parenthesis.
Response: Extra parenthesis has been removed.
I suggest removing the rows from inside the tables to be cleaner.
Response: As suggested by the Reviewer, vertical lines in table 1 have been removed.
I suggest including the data of the calibration curves inside each figure, respectively, and deleting the table (Calibration curves of standards).
Response: As suggested by the Reviewer, one of the tables 2 has been removed. The standard curve equations can be found in Section 2.5.